# Authentication and Chemometric Discrimination of Six Greek PDO Table Olive Varieties through Morphological Characteristics of Their Stones

**DOI:** 10.3390/foods10081829

**Published:** 2021-08-07

**Authors:** Sofia Agriopoulou, Maria Tarapoulouzi, Marie Ampères Bedine Boat, Catherine Rébufa, Nathalie Dupuy, Charis R. Theocharis, Theodoros Varzakas, Sevastianos Roussos, Jacques Artaud

**Affiliations:** 1Department of Food Science and Technology, University of the Peloponnese, Antikalamos, 24100 Kalamata, Greece; t.varzakas@uop.gr; 2Department of Chemistry, University of Cyprus, P.O. Box 20537, Nicosia CY-1678, Cyprus; tarapoulouzi.maria@ucy.ac.cy (M.T.); charis@ucy.ac.cy (C.R.T.); 3Department of Biochemistry, University of Yaounde I, P.O. Box 812, Yaoundé, Cameroon; bedineboat@yahoo.fr; 4Aix Marseille Univ, Avignon Université, CNRS, IRD, IMBE, Campus Saint Jerome, 13007 Marseille, France; c.rebufa@univ-amu.fr (C.R.); nathalie.dupuy@univ-amu.fr (N.D.); sevastianos.roussos@imbe.fr (S.R.); jacques.artaud@univ-amu.fr (J.A.)

**Keywords:** Greek PDO table olive varieties, chemometric analysis, OPLS-DA, discrimination, authenticity, adulteration, geographical origin, quality, safety

## Abstract

Table olives, the number one consumed fermented food in Europe, are widely consumed as they contain many valuable ingredients for health. It is also a food which may be the subject of adulteration, as many different olive varieties with different geographical origin, exist all over the word. In the present study, the image analysis of stones of six main Greek protected designation of origin (PDO) table olive varieties was performed for the control of their authentication and discrimination, with cv. Prasines Chalkidikis, cv. Kalamata Olive, cv. Konservolia Stylidas, cv. Konservolia Amfissis, cv. Throuba Thassos and cv. Throuba Chios being the studied olive varieties. Orthogonal partial least square discriminant analysis (OPLS-DA) was used for discrimination and classification of the six Greek table olive varieties. With a 98.33% of varietal discrimination, the OPLS-DA model proved to be an efficient tool to authentify table olive varieties from their morphological characteristics.

## 1. Introduction

Olive growing is associated with the first steps of human existence, with a history about 5000 years [1], having acquired symbolism associated with peace and friendship as olive branches crowned the winners of the Olympics in Ancient Greece [2]. Especially in Mediterranean countries where the olive tree (*Olea europaea* L. of the family Oleaceae) is considered the most emblematic tree, there exist more than 2000 varieties [3]. From the fruit of the olive trees can produce the well-known olive oil and table olives (also called eating olives) [4,5]. According to International Olive Oil Council (IOOC), “Table olives are the product prepared from the sound fruits of varieties of the cultivated olive tree that are chosen for their production of olives whose volume, shape, flesh-to-stone ratio, fine flesh, taste, firmness and ease of detachment from the stone make them particularly suitable for processing; treated to remove its bitterness and preserved by natural fermentation, or by heat treatment with or without the addition of preservatives; packed with or without covering liquid” [6]. Τhe main producers of table olives in Europe are Spain, Greece, and Italy, other major producers outside the Europe are Egypt, Algeria, Turkey, and Morocco, while emerging producers are Syria, Peru and USA. The word total production exceeded 2.5 million tons in the 2018/2019 season, while the precognition for the 2020/2021 season is to be exceeded 3.0 million tons [7].

Table olives are an important cultural value for societies as a genetic source, displaying a multitude of nutritional characteristics [8]. The quality of table olives is associated with the presence of valuable nutrients and functional bioactive ingredients such as phenolic acids, phenolic alcohols, flavonoids and secoiridoids, and depends on the variety [9]. Consumers highly appreciate olives for their body health as the consumption of olives is associated with many biological activities such as antioxidant, anticarcinogenic and anti-inflammatory and many others pharmaceutical and physiological benefits [5] that allow them to be compared even to those of yogurt [8]. Moreover, olive polyphenols have been used for the prevention of cardiovascular diseases and are highly recommended together with olive oil in the Mediterranean diet [10,11].

The World Catalogue of Olive Varieties which has been compiled under the guidance of IOOC, includes nine Greek olive varieties, namely, *Adramitini*, *Amigdalolia*, *Chalkidiki*, *Kalamon*, *Konservolia*, *Koroneiki*, *Mastoeidis*, *Megaritiki*, and *Valanolia* [12]. Protected designation of origin (PDO) and the protected geographical indication (PGI) are the main designations of origin for agricultural products that are established from European Union (EU) as criteria of authenticity and quality linking these products with origin, geographical indications and traditional specialties [12,13,14,15,16,17,18]. The list in the *World Catalogue of Olive Varieties*, includes indicative olive varieties from all over the world without recording all the existing olive varieties and without all of them being obligatorily marked with a PDO or PGI indication. In addition, these indications refer to products exclusively of the European Union. Until now, ten Greek table olive varieties have been characterized as PDO products. Among them, cv. Prasines Chalkidikis, cv. Kalamata Olives, cv. Konservolia Stylidas, cv. Konservolia Amfissis, cv. Throuba Thassos and cv. Throuba Chios are very famous, and the present study has focused on them. Since tables olives directly come from the tree are not edible, the fermentation process is mandatory, in order to remove oleuropein, which is the main phenolic compound responsible for the bitterness of fresh olives, except from cv. Throuba Thassos and cv. Throuba Chios which have a different debittering process [5].

It is well known that better qualities of olives achieve better prices in the market. As there is plethora of olive varieties with a diversity of morphological and physiological characteristics, the existence of many different qualities is expected [19]. In order to avoid olive adulterations, several discriminant protocols for varietal identification, based on stone, fruit, and leaf data have been used [20].

The authentication of PDO and PGI table olives has been studied the last 15 years in Italy [21], Tunisia [22], Turkey [23], Portugal [24], Greece [25] and Spain [4,26,27]. Several advanced analytical techniques have been used for the study of authentication of table olives, such as high-performance liquid chromatography (HPLC) [23,28], ultra-high-performance liquid chromatography–quadrupole time of flight tandem mass spectrometry (UHPLC-QTOF-MS) [25], gas chromatography-mass spectrometry (GC-MS) [26] and nuclear magnetic resonance spectroscopy (NMR) [21]. Chemometrics is an important science which has been extensively used in food science and authenticity studies to facilitate interpretation of huge load of data, and it provides an easy way to visualize the samples [19,20,29,30,31,32,33,34,35,36].

Characteristics like shape, profile symmetry, front symmetry, basis, apex, mucro, position of maximum transversal width (MTW), number of fibrovascular furrow (NFF), distribution of fibrovascular furrow (DFF), are important and have been used in characterization studies of olive stones [30]. Various standard process of stone processing have been proposed in the literature. In a study by Satorres Martínez et al., three different cleaning methods were applied: a water spray machine, an ultrasonic cleaner and a bleach solution. With the first method, the olive stone was cleaned and part of its texture was damaged. The second method did not have satisfactory results since there were residues of biological material in the texture of the endocarp. Best results were achieved with the last method, the bleach solution. Applying a 5% bleach solution for one hour, there were a complete absence of biological material and no damage appears in the endocarp texture [33]. Bleach solution was also used by Beyaz et al. for the cleaning process of olive stones [20]. Specifically, the olive stones were kept in plastic containers, containing 10% bleach solution, for 15 h and stored at −4 °C to prevent them from cracking because of physiological activity.

To the best of our knowledge limited studies has been reported to investigate the authentication of Greek olive varieties according to the morphological characteristics of their stones [34]. The choice of the six Greek PDO table olive varieties, for the chemometric treatments for varietal identification of olive fruits was based on the coverage of the main cultivated with olives geographical areas of Greece. Thus the cv. Prasines Chalkidikis represent Northern Greece (geographic region of Macedonia), the cv. Konservolia Stylidas, and cv. Konservolia Amfissis represent Central Greece (geographic region of Central Greece), the cv. Kalamata Olive is the most famous all over the Greece and is also characteristic of Southern Greece (geographic region of Peloponnese), and cv. Throuba Thassos and cv. Throuba Chios represent Aegean Sea. The purpose of this work is to discriminate the six Greek PDO table olives, namely cv. Prasines Chalkidikis, cv. Kalamata Olives, cv. Konservolia Stylidas, cv. Konservolia Amfissis, cv. Throuba Thassos and cv. Throuba Chios, regarding the morphological characteristics of their stones and to produce a reliable chemometric model for the authentication of all these table olive varieties.

## 2. Materials and Methods

### 2.1. Olives Sampling

Two sets of olive fruits (perimeter harvested from two olive trees from the same orchard) for each of six Greek PDO table olive varieties, were harvested by hand in the starting of October 2020 from various geographical areas of Greece. These areas are some of the main production areas of PDO table olives in Greece and specifically samples of cv. Prasines Chalkidikis olives were harvested from Chalkidiki (40.20° N, 23.03° E), samples of cv. Throuba Thassos were harvested from Thassos island (40.45° N, 24.35° E), samples of cv. Throuba Chios were harvested from Chios island (38.27° N, 26.07° E), samples of cv. Konservolia Stylidas were harvested from Stylida (38.54° N, 22.37° E), samples of cv. Konservolia Amfissis were harvested from Amfissa (38.28° N, 22.26° E) and samples of cv. Kalamata Olives were harvested from Kalamata (37.05° N, 22.10° E). Figure 1 shows the geographical areas of the analyzed samples of six Greek PDO table olive varieties.

### 2.2. Olive Stone Processing

The olive fruits were transferred to Laboratory of Environmental Biotechnology and Chemometrics, Aix Marseille University, IMBE, and the weight of fresh olives for 60 fruits was measured (30 for each set). The olive fruits were stored at −20 °C for preservation, until the beginning of the analyses. The olive stones are de-fleshed using a procedure developed by Vanloot et al. [11]. Briefly, after thawing, they were placed in hot water for ten minutes and their flesh was removed manually. The olive fruits were brushed to remove all traces of flesh and rinsed with water. The stones were then immersed in hydrogen peroxide for 24 h. They were then rinsed thoroughly to remove all traces of hydrogen peroxide, followed by drying for 48 h at room temperature to obtain a constant weight, which was then measured. This was followed by the storage of the stones in airtight glass bottles until their digital images were obtained. Images were taken from 60 olive stone and for each olive stone, two images (face and profile) were obtained with a high-resolution color camera for 103 character digital processing (Baumer TXD13C) connected on a computer for image processing (Figure 2).

### 2.3. Olive Stone Characteristics

The characterization of the stone parameters was based on the *World Catalogue of Olive Varieties*. The determination of the shape parameters was determined visually on the basis of the different shapes listed in the catalogue. In the *World Catalogue of Olive Varieties* are described with the common glossary the morphological characteristics of tree, inflorescence, leaf, fruit and endocarp (stone) of 139 olive varieties from 23 countries [12]. As it concerns the stone, according to the describing characteristics which are including in the catalogue, it will be very helpful to discriminate the varieties. According to classification there are varieties with low (<0.3 g), medium (0.3–0.45 g) and high (>0.45 g) weight of stones. The shape is characterized as spherical, ovoid, elliptic and elongated when the ratio between the length and width is <1.4, 1.4–1.8, 1.8–2.2 and >2.2, respectively. The symmetry of stone is characterized as symmetric, slightly asymmetric, and asymmetric. The base of the stone which is the part that connects the stone with the peduncle is characterized as truncate, pointed or rounded and apex which is the opposite part of stone it is characterized as pointed or rounded, with or without a mucro. The surface of stone may be smooth, rugose or scabrous [12]. The maximum transversal width can be toward the base, toward the apex or central and the fibrovascular bundles can be deep and abundant. Two positions of the stone, the face and profile, have been used for stone characterization. The first position refers to the maximum symmetry and the second is obtained after rotating 90° from the first. Images were digitized by Visilog v6.7 imaging software from Noesis (Gif sur Yvette, France). Figure 3 shows the detailed characteristics of a stone from Prasines Chalkidikis olive variety.

### 2.4. Application of Chemometrics

SIMCA version 15.0.2 (Umetrics, 907 29 Umeå, Sweden) was used for chemometric analysis. The supervised OPLS-DA procedure was followed to discriminate and classify the observations (samples).

The main limitation of PLS model is its linear nature and it is not applicable for data with non-linear behavior [37,38]. OPLS-DA methods can be applied to visualize variations between sample groups and to define the discriminating performance of the variables. With the use of OPLS-DA it is possible to classify samples according to agricultural practices and predict the origin of unknown samples [39].

Thirteen parameters (variables) were used. Ten of them were related to morphological characteristics: shape, profile symmetry, front symmetry, basis, apex, mucro, MTW, surface, NFF and DFF. Three other parameters were also used related to the weight of the samples, such as average weight of olive fruits, average weight of stones and quantity of olive flesh per olive fruit.

Scaling to unit variance (UV) and mean-centering were used. The samples were discriminated into six classes, namely cv. Kalamata Olive (KO): Class 1, cv. Prasines Chalkidikis (PX): Class 2, cv. Konservolia Stylidas (KS): Class 3, cv. Konservolia Amfissis (KA): Class 4, cv. Throuba Thassos (TT): Class 5, and cv. Throuba Chios (TC): Class 6.

As described in Τarapoulouzi et al. [29] the OPLS-DA model was evaluated here by the determination coefficient, R^2^, reflecting the goodness of fit and the cross-validated correlation coefficient, Q^2^, reflecting the predictive ability of the model. Q^2^ was obtained using the seven-fold leave out procedure (default setting in SIMCA). The ellipse in the plots defines Hotelling’s T2 confidence region, which is a multivariate generalization of Student’s *t* test and provides a 95% confidence interval for the observations. The number of the important components which have been chosen is given with the symbol A, therefore A = 1 + 1 components were used for all the models produced. In addition, internal validation took place with regression models which were validated using CV-ANOVA tables, via comparing F_statistic_ vs. F_critical_ values. F-value is a measure of the size of the effects. The larger this value, the greater the likelihood that the differences between the means are due to something other than chance alone, namely real effects. If the difference between the means is due only to chance, that is, there are no real effects, then the expected value of the F-ratio would be one (1.00). A hypothesis test takes place where the “null hypothesis” indicates that population means of the different appraisers are equal, and “alternate hypothesis” shows that one of the means is not the same. Larger values of F_statistic_ than the F_critical_ indicate that the difference of means of the samples is larger compared to the dispersion of the observations within each sample, and therefore, the null hypothesis should be rejected, and the alternate hypothesis is considered important. In other words, a lower F_statistic_ than the F_critical_ indicates that the variation within the appraisers is greater than the variation between them [40]. The misclassification table was considered important to evaluate the quality of the model, as well as permutation testing was applied (100 permutations) to check the validity and the degree of overfit for the OPLS-DA model.

Validation of the model was tested using sevenfold cross-validation. Therefore, a calibration and a validation set were set up by having 42 and 18 samples, respectively.

## 3. Results

### 3.1. Weight of Olive Stones

In Table 1, the weight of stones of the six Greek PDO table olive varieties are presented. Statistical analyses were performed with SD and these gave the same score scatter plots and classification rates as analyses which did not use them. Generally, the average weight of an olive stone is 18–22% of the olive weight [41]. The average weight of studied fresh olives varied between varieties. In this study, the maximum average weight was observed in the fruits of cv. Prasines Chalkidikis whose average weight was almost ten grams and was twice that of the cv. Kalamata Olive. The cv. Kalamata Olive had the smaller stone and plenty of flesh with the best ratio of olive flesh (90% of the weight of fresh fruit). This feature is extremely interesting to produce olive paste and other olive products from cv. Kalamata Olive. Cv. Throuba Chios had the smaller average weight of fresh olives. As it concerns the weight of olive stones the larger the olive fruit, the larger the stone. The highest average weight of the stone was observed in the cv. Prasines Chalkidikis, followed by the two varieties of Konservolia and the two varieties of Throuba while the smallest average weight of the stone was observed in the cv. Kalamata Olive.

### 3.2. Artificial Visions of Olive Stones

Figure 4 shows the detailed characteristics of face and profile images of olive stones of six analyzed Greek PDO table olive varieties. In Table 2, they are presented the morphological characteristics of stones of six PDO Greek table olive varieties. The images prove that analyzed varieties differ quite except in the case of olive stones of varieties Konservolia. MTW, NFF, DFF and surface are also described in Table 2. All varieties have mucro except from cv. Kalamata Olive. The apexes of cv. Prasines Chalkidikis and cv. Throuba Chios are rounded and the others are pointed. The basis of the stones are pointed for cv. Konservolia Stylidas, cv. Prasines Chalkidikis, cv. Throuba Chios, cv. Throuba Thassos, and cv. Kalamata Olive and only for cv. Konservolia Amfissis is it rounded.

Olive stones from varieties cv. Konservolia Amfissis were morphologically very similar to cv. Konservolia Stylidas. Regarding the shape, cv. Prasines Chalkidikis cv. Kalamata Olive and cv. Throuba Thassos have elongated shape, cv. Konservolia Amfissis, cv. Throuba Chios have elliptic shape and cv. Konservolia Stylidas has an ovoid shape. The profiles of cv. Throuba Chios and cv. Throuba Thassos stones are very asymmetrical, slightly asymmetrical for cv. Prasines Chalkidikis, cv. Konservolia Amfissis, and cv. Konservolia Stylidas stones and asymmetrical for cv. Kalamata Olive stones.

### 3.3. Chemometric Interpretation of the Data by Using OPLS-DA Methods

To tests the validity of the dataset, a calibration and a validation set were set up by having 42 and 18 samples, respectively, as shown in Figure 5. Both scatter plots (a) and (b) were successfully built with R2X(cum) = 0.950, R2Y(cum) = 0.946 and Q2(cum) = 0.933, and R2X(cum) = 0.995, R2Y(cum) = 0.878 and Q2(cum) = 0.789, respectively.

After chemometric interpretation of the data, an overall OPLS-DA model was constructed, as seen in Figure 6. No outlier samples were obtained; thus, all the 60 samples of table olives were distributed in the Hotelling’s T2 ellipse. KO seems to be a very special variety of table olives, as it is the only one which is located at the right part of the ellipse, while the other five varieties (i.e., PX, KS, KA, TT and TC) are located at the center to left part of the ellipse. The values of coefficients R2X(cum) = 0.991, R2Y(cum) = 0.912 and Q2(cum) = 0.855 are all good, since they are all above 0.5, and the difference between R2X(cum) and Q2(cum) is 0.136 which is satisfactory, as it is lower than 0.2–0.3.

Only one sample (PX9) was wrongly classified, as it was located away from the centre of the PX group. It seems that PX9 should belong to TC group, however, chemometric analysis and particularly misclassification table (Table 3) shows that PX9 is located closer to the center of KS group. Calculation of Euclidean distances (results not shown here) confirmed that PX9 is closer to KS group instead of TC. The incorrect classification of PX9 decreases the percentage of correct classification of the PX group to 90%, and the percentage of the overall OPLS-DA model to 98.33%.

By eliminating PX9 from the dataset, the classification rate became 95% as the other samples of PX class were not as closed between them. Thus, it was considered important to keep PX9 in the PX class and in the overall model.

To test the significance and adequacy of the model, the CV-ANOVA, which is considered as the most important test for the evaluation of significance of the developed model, was applied. The CV-ANOVA results show the value of the F_statistic_ and *p*-value and are depicted in Table 4. The model is highly significant, due to the *p*-value of zero. Based on DF = 295, the null hypothesis should be rejected, and the alternate hypothesis is considered important due to that F_statistic_ = 18.9 > F_critical_ = 2.24 for probability level equal to 0.05.

In addition, to validate further the goodness of fit and the predictability of these results, a random permutation test with 100 permutations was employed, as seen in Figure 7. Both R2 (original model) and Q2 (predictive model) located at right and permutated R2 (original model) and Q2 (predictive model) located left while all blue Q2 values to the left and right are lower than the green original R2 values. All the permuted models showed lower R2Y values if compared with the original model’s R2Y value (0.912) and the majority of the Q2 regression lines showed negative intercepts (0.0, −0.688).

## 4. Discussion

The determination of olive variety with this method is very different from the use of precision instruments for material analysis. Not only does it reduce the cost of money and time consumption, but also is more efficient. The identification of the variety of table olives and especially the ones that have been characterized as PDO table olive varieties is required, as the PDO characterization products have higher prices. Variety is a major issue of authenticity and the use of the term PDO can lead to significant falsifications [42]. Since the final product of table olives is a fermented product and different types of table olives can be produced, the methods for determining the variety of origin of fresh olives are completely different from those of table olives, as many changes in the pulp of olives can occur [30]. It is well known that all table olives are fermented in sodium chloride brine [43], through a series of treatments that considerably vary depending on the region and variety [19]. Over the last decade, several studies have been published focusing primarily on reducing sodium chloride content. In this context, modified fermentation brines have been used, in order to satisfy consumers’ demand for healthier table olives, with less sodium chloride [43,44,45,46,47,48,49]. Therefore, different physicochemical characteristics, and sensory and nutritional properties, may arise from the various fermentation procedures, which makes table olive classification difficult.

The study of morphological features of stones and weight measurements of examined Greek varieties constitutes an alternative method that permit us to determine each variety according to their different size, aspect and weight. Besides, using a high-resolution color camera the examined stones are presented in more details, than those, that can be measured with the human naked eye. In addition, the varietal identification achieved with the machine vision system in combination with the chemometric analysis allows fast classification, without the need for human observation and the subsequent errors. In many studies, in addition to the morphological characteristics of the stones, the morphological characteristics of the leaves and fruits have also been used to identify the olive variety. Olive stone information is the most valuable, among other morphological features of a variety, as they are little affected by environmental conditions. Therefore, olive stone characteristics tend to appear similar to olives belonging to the same variety and tend to differ in the opposite case. Martínez et al. approached the problem of varietal identification by feature extraction from the analysis of endocarp images, and then using partial least square-discriminant classifier [33].

This is the first time that the research group studied the authenticity of Greek varieties of table olives, although similar research has been conducted in different countries and varieties by other authors who combined imaging and chemometrics [19,20,30,31,32,34,35]. Esteves da Silva demonstrated the great usefulness of chemometrics in the classification of olive varieties. The morphological characteristics of the endocarp among other characteristics (for example olive fruits, trees, branches, leaves and flowers) were used to classify 22 Portuguese olive varieties. He also managed to demonstrate the similarities between the varieties studied and to show that some characteristics have a greater power of distinction than others [19].

Vanloot et al. achieved the discrimination of five French varieties, namely *Aglandau*, *Bouteillan*, *Lucques*, *Picholine*, *Tanche*, through artificial vision and chemometric analysis of olive stones with 100% of correct classification, working with the data obtained from front and profile pictures [30]. Even if the front and profile parameters are different for the discrimination of the varieties only the picture of profile was sufficient. Image processing techniques of olive fruit, olive leaves, and olive stones, were used for the identification of Turkish olive varieties namely *Sarı ulak*, *Gemlik*, *Edincik su*, *Memecik*, *Eşek zeytini*, *Ayvalık*, *Kilis yağlık*, *Uslu*, *Çilli*, and *Domat* [20], while in another study, image processing techniques with data obtained from the fruits and stones were used for the classification of Spanish olive cultivars, namely *Lechin De Granada*, *Arbequina*, *Picual*, *Verdial De V-M*, *Picudo*, *Hojiblanca* and *Empeltre* [31]. Seven Greek olive varieties, namely *Kalamon*, *Karidolia-Chalkidikis*, *Koroneiki*, *Lianomanako-Tyrou*, *Mastoidis*, *Megaron* and *Throumbolia*, were distinguished according to the morphological parameters of the olive fruit, olive leaves, and olive stones [34]. The study of biometric characteristics of the olive stone was also used to determine the relationships between wild and farmed olives [35].

The analysis regarding geographic origin of the Greek PDO table olive varieties reveals that there is regional clustering. KS and KA were expected to be located closer than the other groups as the locations which have been harvested are nearby. In addition, it can be said that regarding variety species, KS and KA as well as TT and TC were expected to be located “in pairs”, meaning next to each other on the score scatter plot and this is what was observed. These observations show that the varieties from a particular PDO variety can easily be discriminated using the fruit and the stones characteristics. The stones of the KO variety are very characteristic with their very elongated and pointed shape. These stones are very similar to the *Lucques* variety, one of the best and most popular French table olives [50].

Moreover, other authors also stated that image processing alone or coupled with chemometrics can be the best combination in regard to rapidness and ease. Puerto et al. presented a methodology for differentiating olives collected from the ground from those harvested directly from the trees, as the former impoverishes quality of the subsequently produced olive [51]. An automatic inspection system, based on computer vision, was used to classify automatically different batches of olives, before being processed for oil extraction, with a success ratio of 100%. Ponce et al. proposed a non-invasive methodology, in which the classification is carried out uniquely using the morphology of the olive-fruits as distinguishing feature [52]. For this purpose, 2800 fruits belonging to seven different olive varieties, were photographed. It was designed by a procedure, based on image processing and analysis and convolutional neural networks, for developing a set of image classifier. These image classifiers showed a remarkable behaviour in terms of performance, as high rates of accuracy were obtained in general for all of them.

A new methodology, based on computer vision and feature modelling, was proposed by Ponce et al. [53] for automatic counting and individual size and mass estimation of olive-fruits. For its development, a total of 3600 olive-fruits from nine varieties were photographed, stochastically distributing the individuals on the scene, using an ad-hoc designed an imaging chamber. The results from the study indicated relative errors below 0.80% and 1.05% for the estimation of the major and minor axis length for all varieties, respectively.

In a very recent study, an efficient methodology to estimate the maximum/minimum (polar/equatorial) diameter length and mass of olive fruits by means of image analysis was proposed [54]. Different sets of olives from the varieties *Picual* and *Arbequina* were photographed, and an original algorithm based on mathematical morphology and statistical thresholding was developed for segmenting the acquired images. The performance of the models was evaluated on external validation sets, giving relative errors of 0.86% for the major axis, 0.09% for the minor axis and 0.78% for the *Arbequina* variety; analogously, relative errors of 0.03%, 0.29% and 2.39% were annotated for *Picual*.

Diaz et al. dealt the classification of table olive in different quality categories depending on the defects in the surface of the fruits [55]. Learning algorithms that allow the extraction of quality information from batches previously classified by experts have been applied. A colorimetric characterization of the most common defects was performed. An image analysis system was used to segment the parameter set with the olive quality information. The results show that a neural network with a hidden layer can classify olives with more than 90% accuracy.

New effective techniques for automatic detection and classification of external olive fruits defects based on image processing techniques, was presented by Hassan et al. [56]. The proposed techniques can separate between the defected and the healthy olive fruits, and then detect and classify the actual defected area. The proposed techniques are based on texture analysis and the homogeneity texture measure. The results reveal that proposed techniques have the highest accuracy rate among other techniques.

A comparative analysis of the discrimination of pepper (*Capsicum annuum* L.) based on the cross-section and seed textures determined using image processing was developed by Ropelewska and Szwejda-Grzybowska [57]. An effective method based on hyper- spectral imaging combined with a group sparse representation (GSR) classifier for the geographic origin authentication of Yangshan region peaches and to interpret the hyperspectral fingerprint with physiological metabolism using high-performance liquid chromatography (HPLC) analysis was developed by Sun et al. [58].

OPLS-DA method is a very efficient method for discrimination purpose. The NMR-based metabolic profiling tool for the quality assessment of table olives, from the *Konservolia*, *Kalamon* and *Chalkidikis* cultivars from different areas of Greece was used by Beteinakis et al. [59]. Specific biomarkers, related to the classification of olives based on different treatments, cultivars and geographical origin, were identified and OPLS-DA models were built by taking groups in pairs, in order to identify certain markers responsible for the differentiation of cultivars. Moreover, the comparison of similar species in different countries can verify the high discrimination accuracy of OPLS-DA method low-field nuclear magnetic resonance (LF-NMR) in combination with multivariate statistical analysis was used to identify the adulterated Spanish extra virgin olive oil with different rations of soybean oil or corn oil. The multi-blended oil could be 100% classified by OPLS-DA when the adulteration ratio was above 30% [60]. In a very recent study, NMR analysis to avocado oil to differentiate it from other oils including olive oil, was applied by Tang et al. [61]. Avocado oil and olive oil were efficiently classified by OPLS-DA method with an R^2^ of 0.97, and a Q^2^ of 0.91, indicating a very significant model.

This method gave satisfactory results for other agricultural products proving its effectiveness. Becerra-Martınez et al. [62] used NMR spectroscopy supported by principal component analysis PCA or OPLS-DA to differentiate between two Mexican cultivars of chili based on the difference of their metabolites. The authors were able to differentiate the two cultivars using PCA with an R^2^ of 0.936; to better observe differences between groups, OPLS-DA was successfully applied (R^2^ = 0.923). Chung et al. [63] analyzed the multi-element profile of rice samples procured from six different Asian countries using ICP-MS to investigate geographical origin. Rice samples were clearly discriminated through PCA and OPLS-DA as different countries exhibited a different proportion of micro and macro elements.

This work is a pre-study that should be continued in order to increase the database on Greek table olive stones. In future, similar research studies must test more samples per variety and focus on the harvest period. The discrimination of olive varieties can be definitely benefited from the current development of image analysis technology and big data analysis. 

## 5. Conclusions

This research study shows that the morphological features of olives (fruit and endocarp/stone) as well the weight of stones in combination with chemometrics can be discriminated. OPLS-DA proved to be good method for visualizing and interpreting the data. Morphological characteristics of olive stone have enough discrimination capacity to allow to classify the olives. Further research and assessment will take place related to the Greek PDO table olives, and more models can be developed for future predictions related to their quality and authenticity. Ongoing research in the particular field will enlighten the authenticity of the Greek PDO table olives. Τhis preliminary study shows encouraging results and that this visual authentication analysis is easy to implement. It will be more efficient when the image analysis is computerised as planned. This will save time and allow the Greek varieties to be compared with varieties from different geographic origins.

## Figures and Tables

**Figure 1 foods-10-01829-f001:**
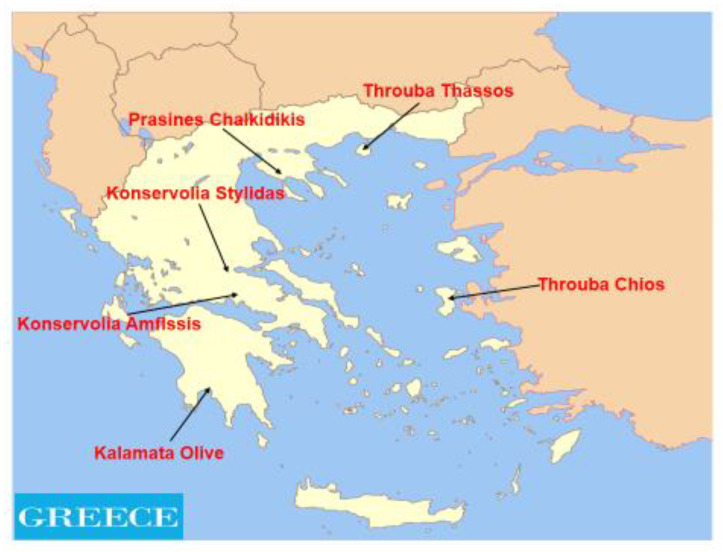
The map of Greece with the geographical areas of the analyzed samples of six Greek PDO table olive varieties.

**Figure 2 foods-10-01829-f002:**
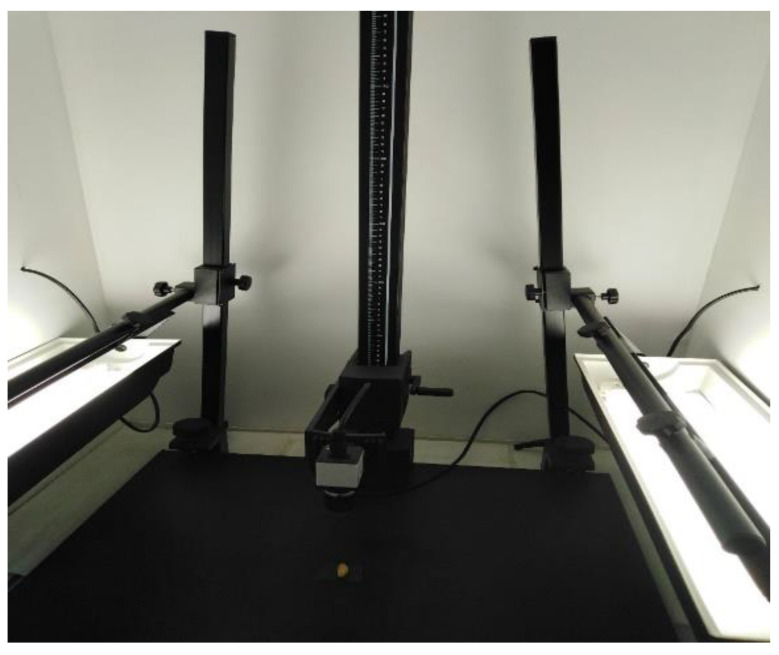
Olive stone image acquisition with a high-resolution color camera (Baumer TXD13C) in center and side lights.

**Figure 3 foods-10-01829-f003:**
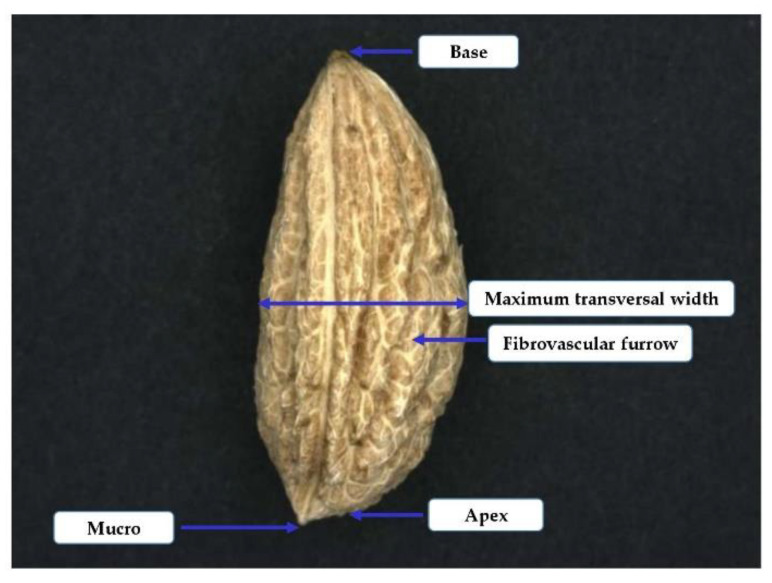
Morphological characteristics of a stone from cv. Prasines Chalkidikis.

**Figure 4 foods-10-01829-f004:**
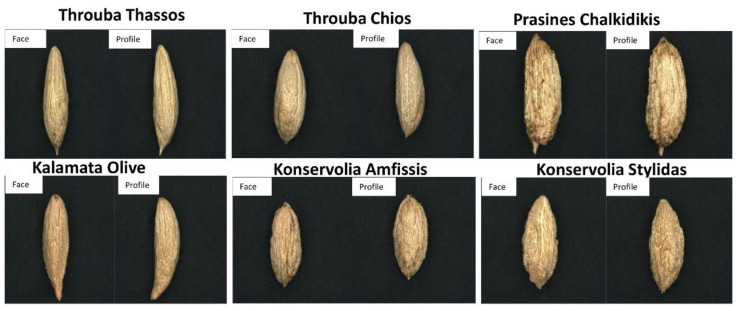
Morphological characteristics of face and profile images of olive stones of six analyzed Greek PDO table olive varieties.

**Figure 5 foods-10-01829-f005:**
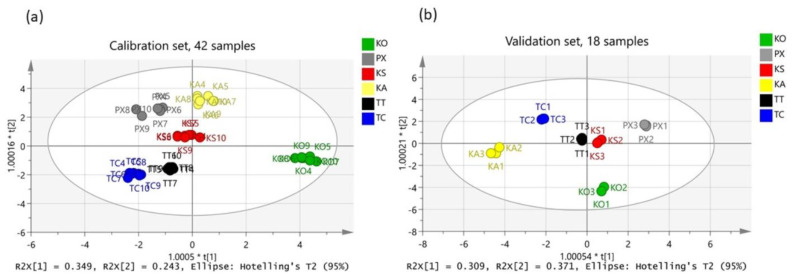
(**a**) OPLS-DA calibration set with R2X(cum) = 0.950, R2Y(cum) = 0.946 and Q2(cum) = 0.933, and (**b**) OPLS-DA validation set with R2X(cum) = 0.995, R2Y(cum) = 0.878 and Q2(cum) = 0.789.

**Figure 6 foods-10-01829-f006:**
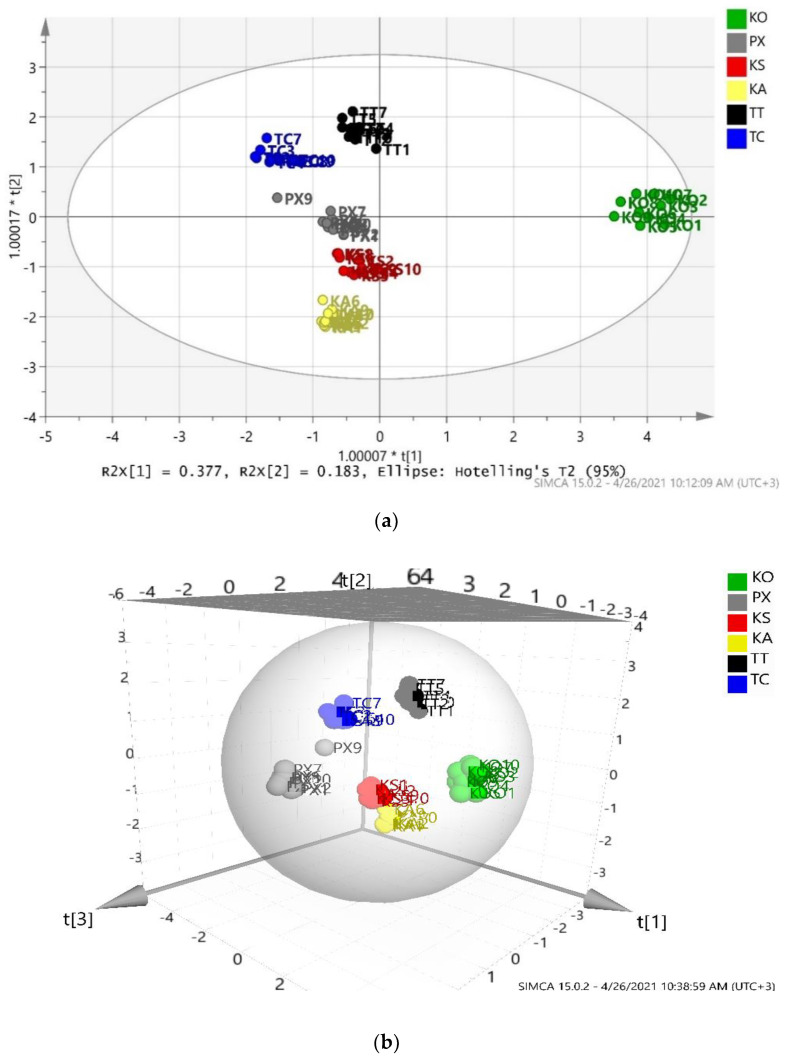
(**a**) The 2D and (**b**) 3D score scatter plots of the overall OPLS-DA model (R2X(cum) = 0.991, R2Y(cum) = 0.912 and Q2(cum) = 0.855.

**Figure 7 foods-10-01829-f007:**
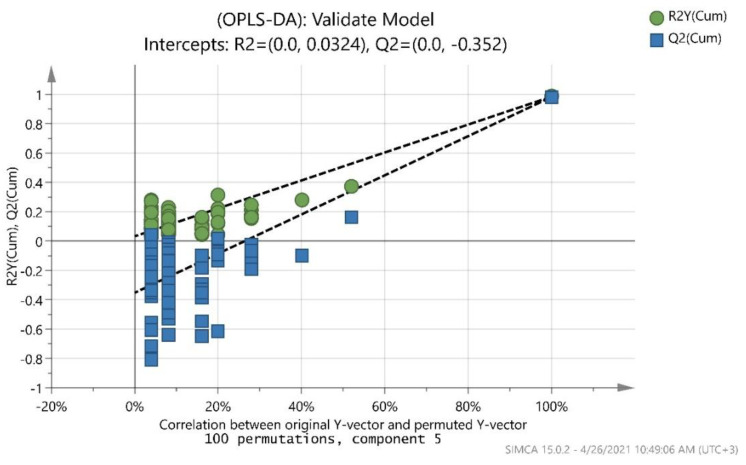
Permutation test of the overall OPLS-DA model took place with 100 permutations.

**Table 1 foods-10-01829-t001:** Weight of 60 stones of six PDO Greek table olive varieties.

Table Olive Variety	Average Weight of Stones (mg) Mean ± SD	Average Weight of Olive Fruits (mg) Mean ± SD	Quantity of Olive Flesh per Olive Fruit (mg) Mean ± SD	Percentage of Flesh (%) Mean ± SD	Percentage of Olive Stone Occupancy (%) Mean ± SD
Kalamata Olive (KO)	489 ± 6	4960 ± 11	4471 ± 7.5	90.2 ± 13	9.8 ± 2
Prasines Chalkidikis (PΧ)	1050 ± 12.6	9710 ± 20.7	8660 ± 45	89.2 ± 24	10.8 ± 4
Konservolia Stylidas (KS)	621 ± 5	5940 ± 9.5	5319 ± 12	89.6 ± 5.5	10.4 ± 3.5
Konservolia Amfissis (KA)	691 ± 5	5950 ± 4.9	5259 ± 9.6	88.4 ± 9	11.6 ± 1.9
Throuba Thassos (TT)	629 ± 5.4	4520 ± 7.8	3891 ± 9	86.1 ± 7	13.9 ± 3
Throuba Chios (TC)	614 ± 9	3030 ± 8.3	2416 ± 10	79.7 ± 10	20.3 ± 6

**Table 2 foods-10-01829-t002:** Morphological characteristics of 60 stones of six PDO Greek table olive varieties.

Table Olive Variety	Shape	Profile Symmetry	Front Symmetry	Basis	Apex	Mucro	MTW ^a^	Surface	NFF ^b^	DFF ^c^
Kalamata Olive (KO)	Elongated	Asymmetrical	Slightly asymmetrical	Pointed	Pointed	Without presence	Middle	Rugged	Weak to middle	Uniform or grouped
Prasines Chalkidikis (PX)	Elongated	Slightly asymmetrical	Symmetrical	Pointed	Rounded	Presence	Middle	Rugged	Middle	Uniform
Konservolia Stylidas (KS)	Ovoid	Slightly asymmetrical	Symmetrical	Pointed	Pointed	Presence	Middle	Rough	Middle	Uniform or grouped
Konservolia Amfissis (KA)	Elliptic	Slightly asymmetrical	Symmetrical	Rounded	Pointed	Presence	Middle	Rough	Middle	Uniform or grouped
Throuba Thassos (TT)	Elongated	Very asymmetrical	Symmetrical to slightly asymmetrical	Pointed	Pointed	Presence	Middle	Smooth to rough	Middle	Uniform
Throuba Chios (TC)	Elliptic	Very asymmetrical	Symmetrical	Pointed or rounded	Rounded	Presence	Middle	Rough	Middle	Uniform

^a^ Position of maximum transversal width. ^b^ Number of fibrovascular furrow. ^c^ Distribution of fibrovascular furrow.

**Table 3 foods-10-01829-t003:** Misclassification table for the overall OPLS-DA model.

	Members	Correct	KO	PX	KS	KA	TT	TC
KO	10	100%	10	0	0	0	0	0
PX	10	90%	0	9	1	0	0	0
KS	10	100%	0	0	10	0	0	0
KA	10	100%	0	0	0	10	0	0
TT	10	100%	0	0	0	0	10	0
TC	10	100%	0	0	0	0	0	10
No class	0		0	0	0	0	0	0
Total	60	98.33%						
Fisher’s prob.	1.1 × 10^−39^							

**Table 4 foods-10-01829-t004:** CV-ANOVA data obtained for the overall OPLS-DA model.

OPLS-DA	SS ^1^	DF ^2^	MS ^3^	F ^4^	p ^5^	SD ^6^
Total corr.	295	295	1			1
Regression	252.2	70	3.6	18.9	0	1.9
Residual	42.7	225	0.19			0.4

^1^ SS = sum of squares, ^2^ DF = degree of freedom, ^3^ MS = mean squares, ^4^ F = F-test calculated value or F_statistic_, ^5^ p = *p*-value of the test, ^6^ SD = standard deviation.

## Data Availability

Not applicable.

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
