# Peer review of "Authentication and Chemometric Discrimination of Six Greek PDO Table Olive Varieties through Morphological Characteristics of Their Stones"

_foods, 2021, doi:10.3390/foods10081829_

Round 1

Reviewer 1 Report

This manuscript (ISSN 2304-8158) studies the authentication of Greek olive varieties according to their stones. They perform a combined analysis using imaging and chemometrics approaches. See below some comments:

- Introduction is a bit confusing since authors explain 9 different varieties in Greece (IOOC) and then they talk about 10 varieties as PDO products. However they didn't study all these 10 varieties and they focused in only 6. Please, improve this paragraph and clarify this fact.

- Table 1: Only averages are provided. Please, add the sd and perform a statistical analysis to group based on the existence of significant differences.

- Discussion is poor and some parts are repetitive (Introduction). 

Reviewer 2 Report

  1. Visually inspect the appearance of different olive stone for analysis. This method is very different from the use of precision instruments for material analysis. Not only reduces the cost of money and time consumption, but also more efficient.
  2. Although this manuscript uses OPLSDA to analyze 13 variables of the six major Greek Protected Designations of Origin (PDO), good results have been obtained.
    But 2 points should be added to the manuscript
    A. Increase the comparison of similar species in different countries to verify that this method is a very efficient method
    B. It is recommended to increase the standard process of stone processing to provide readers with follow-up applications.
  3. t is recommended to expand the image resolution technology after adding more information to build big data

Reviewer 3 Report

The manuscript aims to discriminate six Greek PDO table olive varieties, using a chemometric tool (OPLS-DA) for varietal identification of olive fruits through morphological characteristics of their stones. The study was based on the coverage of the main cultivated with olives geographical areas of Greece. The manuscript is well written and could be a good contribution to Greek olives authentication and quality control assessment. Some remarks that should be detailed in the paper are below.

Specific remarks:

The authors should add more information about how the OPLS-DA calibration model was constructed? How many variables (numbers and names) are used? Why the authors are selected OPLS-DA as a classification technique instead of PLS-DA? What is the difference and what is the significance? Did you think if you eliminate the PX9 sample the classification rate will be 100%? Could you perform a SIMCA classification model (that is more strong to locate outliers)? How did the model be validated, did you think to use cross-validation (e.g. Venetian blinds) procedures to validate it? Why not split your samples between a calibration set (e.g. 80-85%) and a validation set (20-15% of samples)? Why did you not investigate the profile images of olive stones as image data and use them to construct your classification model? I guess the profile images of olive stones may represent a piece of strong information to discriminate between the six olive varieties. 

Round 2

Reviewer 3 Report

The authors are well corrected their paper. The paper was significantly improved as result. Some minor revisions could be performed regarding english style and text editing.